# Factors Affecting the Milk Production Traits and Lactation Curve of the Indigenous River Buffalo Populations in Bangladesh

**DOI:** 10.3390/ani14081248

**Published:** 2024-04-22

**Authors:** Abdullah Ibne Omar, Md. Yousuf Ali Khan, Xin Su, Aashish Dhakal, Shahed Hossain, Mohsin Tarafder Razu, Jingfang Si, Alfredo Pauciullo, Md. Omar Faruque, Yi Zhang

**Affiliations:** 1National Engineering Laboratory for Animal Breeding, Key Laboratory of Animal Genetics, Breeding and Reproduction of Ministry of Agriculture and Rural Affairs, College of Animal Science and Technology, China Agricultural University, Beijing 100193, China; abdullah.ohi86@yahoo.com (A.I.O.); myak.blri@gmail.com (M.Y.A.K.); dhakal_aas1988@yahoo.com (A.D.);; 2Bangladesh Livestock Research Institute, Dhaka 1341, Bangladesh; 3Buffalo Breeding and Developing Farm, Dhaka 1341, Bangladesh; 4Buffalo Development Project (2nd Phase), Department of Livestock Services, Farmgate, Dhaka 1215, Bangladesh; 5Department of Agricultural, Forest and Food Sciences, University of Torino, 10095 Grugliasco, Italy; 6Department of Animal Breeding and Genetics, Bangladesh Agricultural University, Mymensingh 2202, Bangladesh

**Keywords:** Bangladesh, buffalo, lactation yield, calving interval, dry period

## Abstract

**Simple Summary:**

In Bangladesh, buffalo dairy farming is gaining traction for its cost-effectiveness and the nutritional benefits of buffalo milk. This study aims to examine how environmental factors like age, number of past calvings, season of calving, intervals between calvings, and dry periods influence milk yield and lactation patterns. Through monitoring 384 buffaloes from seven populations across different ecological zones, significant impacts of various factors on milk production and reproduction were observed. Findings indicate that such environmental considerations are crucial for enhancing buffalo dairy farming, suggesting that a focus on these aspects could substantially benefit genetic improvement programs.

**Abstract:**

Household buffalo dairy farming is gaining popularity nowadays in Bangladesh because of the outstanding food value of buffalo milk as well as the lower production cost of buffalo compared to cattle. An initiative has recently been taken for the genetic improvement of indigenous dairy buffaloes. The present study was carried out to determine the influence of some environmental factors like age, parity, season of calving, calving interval, dry period on the lactation yield, and lactation curve of indigenous dairy buffaloes of Bangladesh. A total of 384 indigenous dairy buffaloes from the 3rd and 4th parity of seven herds under two different agroecological zones covering four seasons were selected and ear tagged for individual buffalo milk recording. A milk yield of 300 days (MY300d) was calculated following the International Committee for Animal Recording (ICAR) and the data were evaluated using the generalized linear model (GLM). In production traits, the mean of calculated lactation period (CLP), calculated lactation yield (CLY), and milk yield of 300 days (MY300d) of the overall population were 267.28 days, 749.36 kg, and 766.92 kg, respectively, whereas calving interval (CI) and dry period (DP) as reproductive traits were 453.06 days and 185.78 days, respectively. The season of calving, age of buffalo cows, population or herd, agroecological zone, calving interval, and dry period had significant effects on production traits (*p* < 0.05 to *p* < 0.001). The season of calving, level of milk production of 300 days, population, and agroecological zone significantly affected the reproduction traits (*p* < 0.01 to *p* < 0.001). Parity was found to be non-significant for both types of traits. The average peak yield of test day (TD) milk production was highest at TD4 (4.47 kg, 98th day of lactation). The average MY300d of milk production was the highest in the Lalpur buffalo population (1076.13 kg) and the lowest in the buffalo population of Bhola (592.44 kg). The correlations between milk production traits (CLP, CLY, and MY-300d) and reproduction traits (CI and DP) were highly significant (*p* < 0.01 to *p* < 0.001). Positive and high correlation was found within milk traits and reproduction traits, but correlation was negative between milk traits and reproduction traits. Therefore, these non-genetic factors should be considered in the future for any genetic improvement program for indigenous dairy buffaloes in Bangladesh.

## 1. Introduction

Buffalo are distributed on all continents of the world and are found in 77 countries [1]. The world population of buffalo is approximately 208 million. More than 97% of the world’s buffalo are found in Asia; 2% are reared in Africa, particularly in Egypt; 0.7% are in South America, and fewer than 0.2% are in Australia and Europe [2]. Bangladesh possesses approximately 0.64 million buffaloes [3], but no specific breeds have been recognized so far [4,5,6]. Cattle and buffalo are the most important sources of milk, meat, and draught power in Bangladesh. Farm mechanization displaced animal power to the extent of 95 to 100% [7]. Therefore, the utility of buffalo as draught animals has shifted from draught to dairy animals. Dairy buffaloes are reared in the attached chars and off-shore islands under semi-intensive and extensive management system [8]. Household buffalo dairy farming is gaining popularity nowadays for the outstanding food value of buffalo milk as well as the lower production cost of buffalo compared to cattle [8,9]. However, there are two problems with setting up a buffalo dairy farm. The first is the low number of female buffaloes and the second is the low milk yield of local buffalo cows [8,10,11]. An initiative has recently been taken for the genetic improvement of indigenous dairy buffaloes here [9].

To increase profitability of dairy buffaloes, it is necessary to know the factors affecting their milk production and reproduction ability [12]. Variation in milk production is a regular phenomenon in all milking animals. Such variation occurs due to (i) genetic factors controlled by the genetic make-up of the animal, and (ii) environmental factors such as age, parity, season of calving, calving interval, dry period, and nutrition status. The effects of non-genetic factors on the milk yield of dairy buffaloes have been reported in a number of investigations both in developing [13,14,15] and advanced countries [16,17]. Those investigators [13,14,15,16,17] reported that total milk yield, lactation length, and dry period are affected by parity, season of calving, calving interval, dry period, and nutrition status. Thus, non-genetic factors like parity, season of calving, and dry period all affect the milk production and reproductive performances of dairy buffaloes. Analysis of the literature revealed that observations on the effect of non-genetic factors on the first lactation and lifetime performance of Bangladeshi indigenous dairy buffaloes are scarce. A lone study on the effect of the environmental factors has been conducted. Parity was observed to influence lactation length and calving interval among indigenous buffaloes in the southern region of Bangladesh, as indicated by data derived from farmer interviews [18].

Milk production is one of the main sources of income for dairy farms and recording milk yield is essential for dairy herds to easily understand and create their lactation curve, which refers to a graph showing the relationship between milk yield and time after calving to drying off [19]. The lactation curve is a mathematical framework model that depicts the variation in milk yield during lactation, and the knowledge of the lactation curve makes it possible to estimate the total milk yield from test days in the lactation process [20,21]. Various mathematical models have been developed for modelling the lactation curve [21,22]. The incomplete gamma function (Wood model) [23], one of the most widely used functions, can generate the standard shape of the lactation curve and is particularly useful for modelling average curves of homogeneous groups of animals [22]. Another group of models, including the Ali–Schaeffer model [24], the Wilmink model [25], and the Legendre orthogonal polynomials [26,27], is suitable for modelling the individual curve shapes [22].

Lactation curves of buffalo have been presented by a number of investigators [28,29,30]. In a study of the factors affecting the shape of the lactation curve in Nili-Ravi buffaloes in Pakistan [28], the variations in shape of the lactation curve were observed due to parity, season of calving, and lactation length. There are different models to describe the lactation curve of dairy buffaloes. Non-linear models were also used to fit the lactation curves for milk yield and composition of buffaloes [29]. In addition, the orthogonal polynomials were evaluated for test day milk yield recording of Murrah buffaloes [30]. However, there is no information on the test day milk recording as well as lactation curve of indigenous dairy buffaloes in Bangladesh.

The present study was carried out to determine the influence of some environmental factors like age, parity, season of calving, calving interval, dry period on the lactation yield, and lactation curve of indigenous dairy buffaloes of Bangladesh based on animal recording of farmer herds. It is envisaged that the information so generated would be helpful in formulating a future improvement program for indigenous dairy buffaloes in Bangladesh.

## 2. Materials and Methods

### 2.1. Ethical Statement

All the experiments on animals were conducted according to the rules and regulations and guidelines of animal welfare approved by the standing committee of the Bangladesh livestock department. No animals were stressed or harmed during the milk production and reproductive data collection. Data were collected only with the explicit agreement and permission of the owners, thus obviating the requirement for an ethical statement in this study.

### 2.2. Locations of Experiments

Two ecological zones were selected where indigenous dairy buffaloes are found in Bangladesh. These were the Flood Fed Area (FFA) and Coastal Area (CA). FFA included the Padma river basin and the Brahmanputra river basin. Experimental dairy buffaloes were Godagari, Paba, and Lalpur in the Padma river basin; and Trishal, Jamalpur, and Madargonj in the Brahmanputra river basin. Bhola was in the coastal area. The geographical locations of the experimental sites were 24°39′ N and 90°24′ E, 24°55′ N and 89°57′ E, 24°53.5′ N and 89°45′ E, 24°10′ N and 88°58′ E, 24°26′ N and 88°37′ E, 24°28′ N and 88°19′ E, and 22°41′ N and 90°38′ E for Trishal, Jamalpur, Madargonj, Lalpur, Poba, Godagari, and Bhola, respectively (Figure 1 and Figure 2). The climate was generally hot, humid, and sub-tropical in nature and nearly the same for all locations. The mean annual maximum and minimum temperatures were 33 °C and 24.7 °C, respectively, and the mean relative humidity ranged from 69.2% to 76.2%.

### 2.3. Experimental Animals

A total of 384 indigenous dairy buffaloes in the 3rd or 4th parity were selected from seven different populations under two different ecological zones, which were the Flood Fed Area (FFA) and Coastal Area (CA). Two areas were selected from the FFA, in which the first area was the Padma River basin, where buffaloes were selected from Godagari (69), Paba (40), and Lalpur (40). The second area was the Brahmanputra river basin, where animals were selected from Trishal (25), Jamalpur Sadar (40), and Madargonj (70). Bhola was located in CA, from where 100 buffalo were selected.

### 2.4. Management of Experimental Animals

All the experimental buffaloes were managed under a semi-intensive system. In this system, buffaloes were kept confined to homesteads under open air during the night. In the morning, milking was done through the suckling method, i.e., calves were allowed to suckle the dam during milking. Then, buffalo cows and calves were allowed to graze in a natural pasture in the nearby attached chars of the river or offshore inlands of the Bay of Bengal. The buffaloes were driven back to homestead or a fixed location on the attached char or offshore island in the evening. The farmers possessing better milch buffaloes in the Padma–Brahmanputra river basin (producing more than 4 kg of milk) provided wheat bran to milking buffalo cows from 1.0 to 2.0 kg/head/day. Some farmers practiced milking twice a day. All the experimental buffaloes were marked with ear tags and a unique number was given to each animal. Routine deworming was done by using A-mectin plus Vet (Ivermectin BP and Clorsulon USP) injection for 1mL/50 kg BW thrice a year and vaccination was carried out with 2 mL/animal against *Haemorrhagic septicaemia* twice a year.

### 2.5. Milk Recording

In the absence of any animal recording of economic traits in dairy animals (both buffalo and cattle) of Bangladesh, we selected some traits; namely, calculated lactation period, calculated lactation yield, dry period, and calving interval as important economic traits for dairy buffaloes in Bangladesh. Likewise, in the absence of any animal recording, we used ICAR Standards for recording and sampling intervals [31] and ICAR Guidelines for Computing of Accumulated Lactation Yield: Computing Lactation Yield [32]. To make the recording system easy for the farmer, we took 11 records with a constant 28-day interval for the 2nd to 10th records and a 14-day interval for the 1st and 11th records. It may be mentioned here that a 305-day lactation period is considered for Indian buffalo breeds. Since Bangladesh indigenous dairy buffaloes are of Indian origin, we set a 300-day lactation period instead of 270 days as the total lactation period. We selected the buffaloes as per desired genotype and marked the buffaloes with leaser-printed ear tags. Farmers were well trained about animal recording and maintained a record sheet for each individual cow for milk records, date of calving, date of drying off, etc. Lactating buffalo cows were milked manually around 7.00 a.m. in the morning and in the evening at 6.00 p.m. (for those that were milked twice a day) during the lactation period. The milk yield was measured using a digital measuring scale with a sensitivity of at least 100 g in each milking. A total of 11 test day milk yields (TDMY) for 384 individual buffalo cows were taken. The milk yield of 300 days was calculated following the test interval method [33] as described in ICAR Guidelines [32].

### 2.6. Data Arrangement

The location of the experimental site (agroecological zone), buffalo population, animal identification number, parity of buffalo, calculated lactation period, calculated lactation yield, milk yield in 300 days, dry period, season of calving, and calving interval were recorded. The data were grouped according to the agroecological zone (AEZ), season of calving (SOC), age of lactating buffalo, population, parity, level of milk production (LMP), calving interval (CI), and dry period (DP) to quantify their effect on productive traits (calculated lactation period—CLP; calculated lactation yield—CLY; milk yield in 300 days—MY300d) and reproductive traits (calving interval—CI; dry period—DP). Seasons were classified into 4 categories: rainy season (June–August), autumn season (September–November), winter season (December–February), and summer season (March–May). DP were divided into 3 groups (<90 days, 90–150 days, and >150 days) and CI were divided in to 2 groups (<15 months and >15 months) for the production traits of 384 buffaloes.

### 2.7. Statistical Analysis

The relationship between non-genetic factors with production traits (CLP, CLY, and MY300d) and reproduction traits (CI and DP) were evaluated by the Generalized Linear Model (GLM) using the “Agricolae” package [34] of R software, version 4.3.0 [35]. The Duncan multiple range test was used [36], and Pearson’s correlation analysis was performed between production traits (CLP, CLY, and MY300d) and reproduction traits (CI and DP) [37]. The following statistical models were used to find out the effects of non-genetic factors on the production traits (Model 1) and reproduction traits (Model 2):

Model 1:$$Y_{ijkmopq}=\mu +S_{i}+P_{j}+A_{k}+{CI}_{m}+{PO}_{o}+{DP}_{p}+Z_{q}+e_{ijkmopq}$$

Model 2:$$Y_{ijnoq}=\mu +S_{i}+P_{j}+L_{n}+{PO}_{o}+Z_{q}+e_{ijnoq}$$
where *Y_ijkmnop_* and *Y_ijkmn_* were the observed value of production traits of CLP, CLY, and MY300d (Model 1) and reproduction traits of CI and DP (Model 2); *μ* was the overall population mean for the traits. Both models had common parameters represented by the season of calving (*S*), parity (*P*), population (*PO*), and agroecological zone (*Z*), in which *S* was the effect of the *i*th season of calving (where, *i* = 1, 2, 3, and 4; in which 1 = Rainy, 2 = Autumn, 3 = Winter, and 4 = Summer season, respectively); *P* was the effect of the *j*th parity (where *j* = 3rd and 4th); *A* was the effect of the *k*th age of buffalo cow (where, *k* = 7, 8, and 9 years); *CI* was the effect of the *m*th calving interval (where *m* = 1 and 2); *PO* was the effect of the *o*th population (where *o* = 1, 2, 3, 4, 5, 6, and 7; in which 1 = Poba, 2 = Godagari, 3 = Lalpur, 4 = Madargonj, 5 = Jamalpur, 6 = Trishal, and 7 = Bhola population, respectively); *Z* was the effect of the *q*th agroecological zone (where, *q* = 1 and 2; in which 1 = Flood Fed Area and 2 = Coastal Area). *DP* was the effect of the *p*th dry period (where *p* = 1, 2, and 3; in which 1 = less than 90 days, 2 = 90–150 days, and 3 = more than 150 days) and *L* was the effect of the *n*th level of the milk production of 300 days, (where *n* = 1, 2, and 3, in which 1 = less than 600 kg, 2 = 600–1000 kg, and 3 = more than 1000 kg) for Model 2 ; *e**_ijkmopq_*** and *e**_ijnoq_*** stand for the random residual error. The least-square means with the standard error of mean w performed using Tukey correction using a statistical package of R software (https://cran.r-project.org/web/packages/lsmeans/ accessed on 15 May 2023). The Pearson correlation coefficient was calculated among CLP, CLY, MY300d, CI, and DP.

### 2.8. Estimation of Lactation Curve and Goodness of Fit

To describe the lactation curve of the indigenous river buffalo populations of Bangladesh, we used the test day (TD) milk records to evaluate the lactation curve parameters. The Wood lactation curve model [23], which is widely popular and used to explore the lactation curve in dairy species, was used. In each buffalo herd’s daily milk yield data, records with a daily milk yield of zero were removed. The lactation curve for the herd was then fitted using the Wood model, and the goodness of fit of the model was evaluated by the coefficient of determination (R^2^ = 1 − RSS/TSS, where RSS and TSS are sum of squared residuals and total sum of squares, respectively) and the root mean square error (RMSE) to ensure the explanatory power of the model [38,39]. The Wood model is as described by the formula:$$Y_{t}=at^{b}e^{-ct}$$
where t represents the day of lactation; $Y_{t}$ represents the daily milk yield for the lactation days; a represents the lactation potential of the buffalo, b represents the rate of decline in the lactation curve, and c represents the rate at which the curve reaches its peak; e represents the mathematic constant. The model parameters were used to estimate the peak lactation day ($t_{m}$) and peak milk yield ($y_{m}$) as described by the following formulas:$$t_{m}=\frac{b}{c},\;\;{\;y}_{m}=a{\left(\frac{b}{c}\right)}^{b}e^{-b}$$

The “nlme” package was used for statistical analysis [40].

## 3. Results

### 3.1. Factors Affecting the Milk Production Traits

The effects of non-genetic factors such as the season of calving (SOC), age, population (herd), agroecological zone (AEZ), calving interval (CI), dry period (DP), and parity on the milk production traits, namely, the calculated lactation period (CLP), calculated lactation yield (CLY), and milk yield in 300 days (MY300d), were calculated and presented in Table 1.

SOC, age, population, AEZ, CI, and DP significantly (*p* < 0.05) or highly significantly (*p* < 0.001) affected CLP. The least-squares mean of CLP was highest in the autumn season (268.81 days) followed by the winter season and rainy season. The mean of CLP was the highest in the summer season but the sample size was very small (Table 1). The CLP was found highest at the age of 7 years (270.51 days) and lowest at the age of 9 years (Table 1). The mean of CLP was found to differ significantly (*p* <0.001) among the populations, where the shortest period was observed in the Bhola population (256.60 days) and the longest period was observed in the Lalpur population (279.54 days). For the rest of the populations, the range of CLP was observed from 261.50 to 269.53 days. For the AEZ, the buffaloes of FFA had a higher lactation period than the buffaloes of CA. The CLP was higher in the group of buffaloes, which had a CI less than 15 months (275.32 days) and a DP less than 90 days (292.11 days). The CLP was the lowest in the group of buffaloes whose DP was more than 150 days (263.49 days) (Table 1).

Age, population, AEZ, CI, and DP had a highly significant effect (*p* < 0.001) on CLY and MY300d. SOC also had a significantly high effect (*p* < 0.01) on CLY and MY300d (Table 1). Considering the number of buffalo calved in different seasons, most of the progenies were obtained in rainy and autumn seasons. CLY and MY300d values were higher (747.81 kg and 764.39 kg, respectively) for buffaloes that calved in the autumn season, while lower values were obtained for buffaloes that calved in the rainy season (741.39 kg and 759.61 kg, respectively). Among the age variation of animals, the higher average CLY and MY300d (812.59 kg and 830.22 kg, respectively) were observed more at a younger age (7 years) than the older age (9 years) and differed significantly (*p* < 0.05) among the age groups. There were significant differences (*p* < 0.05) among the populations for CLY and MY300d, where the highest and lowest values of CLY and MY300d were recorded for the population of Lalpur (1046.33 kg and 1076.13 kg, respectively) and Bhola (576.59 kg and 592.44 kg, respectively). The effect of AEZ, CI, and DP was also significant (*p* < 0.001) on CLY and MY300d and the highest values were obtained (810.19 kg and 828.36 kg), (923.63 kg and 942.59 kg), and (1649.22 kg and 1671.24 kg) for the buffaloes of FFA of AEZ, with less than 15 months of CI and less than 90 days of DP, respectively.

### 3.2. Factors Affecting the Reproduction Traits

The least-squares means for the effect of parity, SOC, level of milk production in 300 days (LMP-300d), population, and AEZ on the reproductive traits of CI and DP were calculated and have been presented in Table 2. The calving interval (CI) and dry period (DP) were affected significantly (*p* < 0.01 to *p* < 0.001) by SOC, LMP-300d, population, and AEZ.

### 3.3. Descriptive Statistics of Test Day (TD) Milk Yield Records

The TD milk production data are presented in Table 3. The average milk production increased from TD1 (1.51 kg, 14th day of lactation) to a pick yield at TD4 (4.47 kg, 98th day of lactation) and subsequently declined until the end of lactation at TD10 (1.15 kg, 266th day of lactation) (Figure 3). Lalpur and Madargonj are the most productive populations while Poba, Godagari, Trishal, and Bhola produced less milk across the lactation (Figure 3). There was a large variation in milk yield between individuals within the population as well as between the populations, which was reflected by the coefficient of variation (CV) (Table 1).

### 3.4. Correlation between Production Traits and Reproduction Traits

The correlation coefficients between production traits (CLP, CLY, and MY300d) and reproductive traits (CI and DP) exhibited significant negative associations (*p* < 0.01 to *p* < 0.001) with each other, as shown in Table 4. Among the milk production traits (CLP, CLY, and MY300d) and reproduction traits (CI and DP), the correlation coefficients of CLP were positive for CLY (*r* = 0.77, *p* < 0.001) and MY300d (*r* = 0.76, *p* < 0.001), but they were significant and negatively correlated with CI (*r* = −0.67, *p* < 0.01) and DP (*r* = −0.81, *p* < 0.001) in overall population presented in Table 4. Similarly, CLY and MY300d were also highly correlated with each other; CLY showed a very high correlation (*r* = 0.98, *p* < 0.001) with MY300d, but it was negative and highly significant (*p* < 0.001) for CI and DP traits in Bangladeshi buffalo populations. On the other hand, the CI and DP were positively correlated with each other with very high significant values (*r* = 0.98, *p* < 0.001) but both were negatively correlated with the milk production traits of CLP, CLY, and MY300d (*r* = −0.67, −0.73, and −0.73 with CI, and *r* = −0.81, −0.79, and −0.78 with DP, respectively) with a significant level between *p* < 0.01 and *p* < 0.001 in this study (Table 4).

### 3.5. Lactation Curve in Different Buffalo Populations

The estimation of lactation curve parameters (i.e., a, b, and c) of incomplete gamma function (Wood model) and the peak lactation days (*t_m_*) and peak milk yield (*y_m_*) by utilizing the test day (TD) milk record data of different buffalo populations of Bangladesh has been presented in Table 5 and the lactation curve drawn has been shown in Figure 4.

Among the buffalo populations, the Lalpur buffalo population showed the highest daily milk yield followed by the Madargonj population. The peak lactation day of the Lalpur buffalo population was at an observed height on the 73th day among populations and then declined up to day 300 of milking. Similarly, daily milk yield increased gradually in the buffalo population of Madargonj, Jamalpur, Godagari, Trishal, and Poba, respectively (Figure 4), and peak lactation day was observed on the 64th day, 65th day, 59th day, 64th day, and 56th day of milking, respectively (between the TD2 and TD3) (Table 5). The buffalo cows of Bhola showed the lowest daily milk yield with a comparatively higher period to reach peak (excepted the Lalpur buffalo population of the 73th day) lactation day (68th day between TD2 and TD3) among all buffalo populations. The coefficient of determination (R^2^) rate was the lowest (0.396) in the buffalo population of Lalpur and the highest (0.659) was observed in the buffalo population of Trishal, as shown in Table 5.

## 4. Discussion

Information on the performance of lactating buffaloes in Bangladesh is scanty. Few pieces of literature are available on the performances of Bangladeshi indigenous dairy buffaloes based on farmers’ interview data [41,42,43]. To our knowledge, this is the first report on the effect of non-genetic factors on lactation period, lactation yield, calving interval, and lactation curve of indigenous dairy buffalo populations across different regions of Bangladesh based on recorded data.

In the present study, age, SOC, population, AEZ, CI, and DP had a high significant effect on CLP, CLY, and MY300d. Previous studies showed that SOC had a mild effect on lactation yield (LY) and lactation period (LP) in Murrah buffalo cows [44,45,46] and a significant effect on the lactation period in Nili-Ravi buffalo [14]. Several investigators [47,48,49,50] reported that SOC significantly affected (*p* < 0.01) LY in Murrah buffalo cows, which supports our present findings. Meanwhile, a significant effect of SOC on MY300d was found in Murrah buffalo [48,49,51] and in Nili-Ravi buffalo [52]. In our findings, the Bangladeshi buffalo showed the higher milk production that calved in autumn, while the rainy-calving buffalo had the lower milk production. In addition, maximum milk production was observed in winter-calving buffalo followed by the spring-calving, autumn-calving, and summer-calving buffalo in Nili-Ravi cows [53], Egyptian buffalo cows [37], and Murrah cows [54,55], respectively. This variation might be due to feed and nutritional effect. In the FFA of Bangladesh, natural pasture is inundated by flood water in rainy season and ruminant livestock, including buffalo, suffer from severe feed deficiency. At the beginning of autumn, locally cultivated legumes, sugarcane, and naturally grown grass are available for buffalo feeding and continue up to the end of winter. In the present study, CLP, CLY, and MY300d were highly significant (*p* < 0.001) among the different populations; CLP, CLY, and MY300d was the highest in the Lalpur population, which grazed in the Padma river basin, and the lowest values were observed in the population of the Bhola region (Table 1). Bangladeshi dairy buffalo are reared in two AEZs: the Flood Fed Area (FFA), which included Padma river basin and Brahmaputra river basin and possesses improved river type buffaloes, and the Costal Area (CA), which includes a saline area; most of the primitive river buffalo were found in that area.

The overall mean of calving interval (CI) in our current study was 453.04 days, which was in line with the findings for Murrah buffalo in India and Nepal [56,57]. However, the CI was found higher for Nili-Ravi, Murrah, and Surti buffaloes in Sri Lanka than our present study [58]. Our result showed that buffaloes producing higher milk yield and longer lactation period had shorter period of CI (less than 15 months). This result is inconsistent with some reports that Egyptian buffaloes with higher milk yield had the longer period of CI [37,59]. In the present study, CI was significantly affected by the season of calving (*p* < 0.01), which was consistent with some other reports on Murrah and Egyptian buffaloes [15,56,60]. LMP-300d, population, and agroecological zone (AEZ) significantly affected CI in our current study (Table 2). In the present study, parity did not affect CI. However, previous investigations reported that CI was affected significantly (*p* < 0.01) by parity in Egyptian, Daira, and Murrah buffaloes [15,57]. This might be due to considering only 3rd and 4th parity in our study instead of considering 1st to 6th parity. In the current study, highly significant (*p* < 0.001) effect of the dry period (DP) on CLP, CLY, and MY300d was observed in Bangladeshi buffalo cows. We found higher mean of CLP, CLY, and MY300d in the buffalo cows whose dry period was less than 90 days. This is consistent with the findings in Egyptian buffaloes [37] and Murrah buffaloes [61]. They reported highest milk yield from buffalo cows whose DP was less than 90 days. Table 2 reveals that the season of calving had a significant effect (*p* < 0.01) on the dry period. Comparable results were also reported in Murrah buffaloes [51,62]. Buffaloes calving in summer season had the shortest dry period (91.00 ± 3.00 days), whereas those calving in rainy season had the longest dry period (201.71 ± 4.18 days), which agreed to the finding in Murrah buffaloes [56]. A significant effect on DP (*p* < 0.01) for MY300d was observed in the current study where DP decreased with the increasing milk production (Table 2). Population had a more significant effect (*p* < 0.001) on dry period. A similar result was also found for AEZ. This type of variation in CI and DP may be due to the fodder and feed availability as well as the heat expression by season and the variation of the local breeds.

The test day (TD) milk production data (Table 3) indicates that the average milk production increased from TD1 (14th day of lactation) to a peak yield of TD4 (98th day of lactation) and subsequently declined until the end of lactation on TD11. The pattern was consistent with the previous findings in Brazilian buffalo cows [63] and in Colombian buffalo cows [64]. In addition, this study showed significant positive correlations among production traits (CLP, CLY, and MY300d) as reported in Egyptian buffaloes [37] and Nili Ravi buffaloes [65].

To compare the lactation curves for different Bangladeshi buffalo populations, we used the Wood lactation model. In our current study, the highest peak of test day milk yield was recorded in Madargonj population and the lowest peak of monthly test day milk yield in Bhola population. The coefficient of determination (R^2^), an indication of goodness of model fit, varied between 0.396 and 0.659, which are lower than those reported in Anatolian buffalo (0.760) [19], Murrah and Surti buffaloes in Sri Lanka (0.813) [66], Indian Murrah buffalo (0.931) [67], and Iranian buffalo (0.841~0.850) [29]. In this study, the buffaloes were reared under a semi-intensive system by smallholder farmers, which could explain the large residual for modelling. The estimated parameter a of the Wood function ranged from 0.386 to 0.527 for the seven Bangladesh buffalo populations, which are comparable to Murrah and Surti buffaloes in Sri Lanka (0.573) [66], but lower than most previous studies [19,29,39,67,68]. As this parameter represents the initial milk yield of lactation [23], the small value indicated the lower milk production performance of Bangladesh buffaloes, as also supported by the low peak yield (2.850~4.269 kg/day) (Table 5). The estimated peak times (56.878~73.083 days) of Bangladesh buffaloes, however, are generally consistent with the previous findings in different buffalo breeds [19,20,39]. By considering the reports on the different buffalo breeds and current result, we assumed that the Wood lactation model is appropriate for explaining the lactation curves based upon milk yield records for the indigenous river buffalo populations in Bangladesh.

## 5. Conclusions

It can be concluded that the season of calving, calving interval, age of lactating cows, dry period, the population of buffaloes (herd), and agroecological zone had a significant effect on milk production traits. Similarly, season of calving, level of milk production in 300 days, population, and agroecological zone had also significant effect on reproductive performance of Bangladeshi dairy buffaloes. However, parity had no effect on the milk production traits as well as reproductive traits of those buffaloes. Therefore, both genetic and non-genetic factors should be considered in future for any improvement program of indigenous dairy buffaloes in Bangladesh.

## Figures and Tables

**Figure 1 animals-14-01248-f001:**
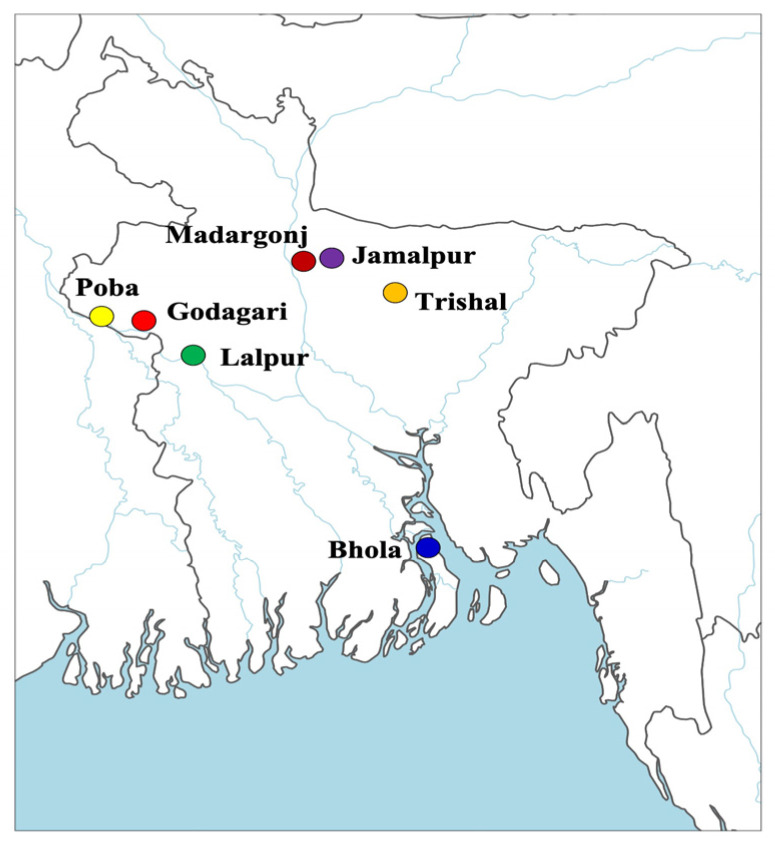
Geographical location of the experimental sites for the milking records of Bangladeshi river buffalo populations.

**Figure 2 animals-14-01248-f002:**
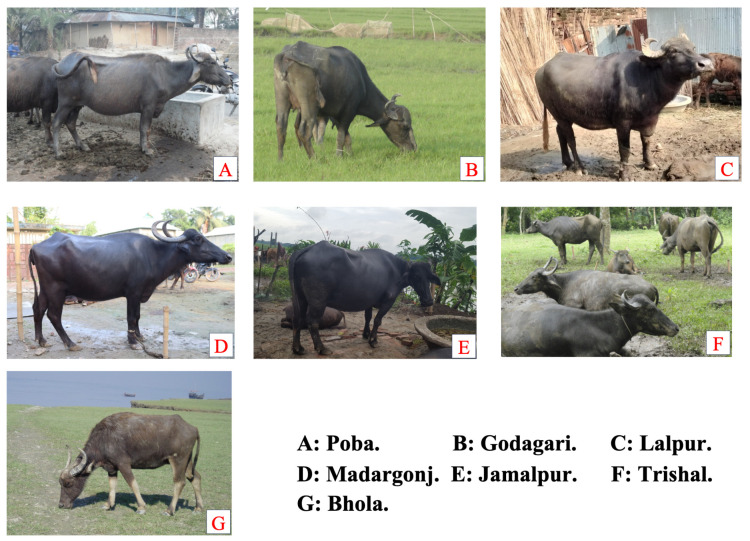
Seven populations of indigenous dairy buffalo in Bangladesh.

**Figure 3 animals-14-01248-f003:**
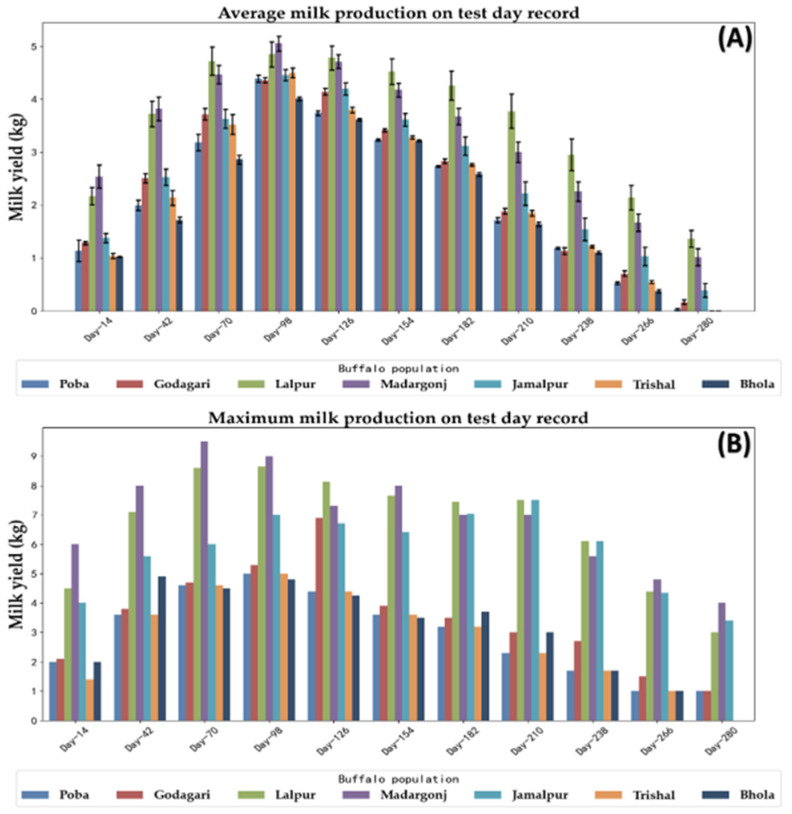
(**A**) Average milk production and (**B**) maximum milk production on test day records for different buffalo populations of Bangladesh under village conditions.

**Figure 4 animals-14-01248-f004:**
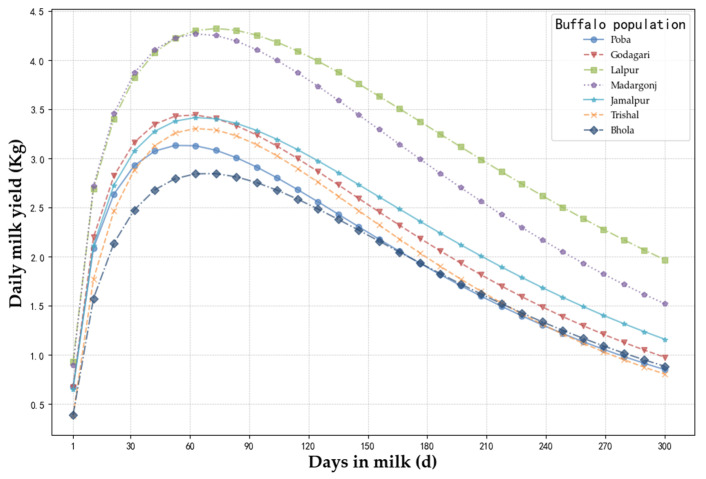
Lactation curves of milk yield for seven populations based on the Wood model.

**Table 1 animals-14-01248-t001:** Least-squares mean (LSM) of factors affecting the milk production traits (CLP, CLY, and MY300d) of indigenous river buffalo populations of Bangladesh.

Factors ^1^	N ^2^	CLP, Days			CLY, Kg			Milk Yield of 300 Days, Kg		
		Max	Min	Mean ± SEM ^4^	Max	Min	Mean ± SEM ^4^	Max	Min	Mean ± SEM ^4^
Overall	384	298	240	267.28 ± 0.68	1888	446	749.36 ± 15.08	1902.0	484.4	766.92 ± 15.32
Parity		NS ^3^			NS			NS		
3rd	172	298	240	267.22 ± 0.99 ^a^	1790	497	752.65 ± 23.13 ^a^	1800.3	508.3	768.83 ± 23.45 ^a^
4th	212	296	241	267.32 ± 0.93 ^a^	1888	446	746.68 ± 19.92 ^a^	1902.0	484.4	765.38 ± 20.24 ^a^
SOC		*			**			**		
Rainy	185	298	240	265.48 ± 1.05 ^a^	1888	446	741.39 ± 23.60 ^b^	1902.0	484.4	759.61 ± 23.77 ^b^
Autumn	195	296	242	268.81 ± 0.87 ^a^	1780	476	747.81 ± 18.60 ^b^	1798.8	525.1	764.39 ± 19.02 ^b^
Winter	2	280	256	268.00 ± 2.00 ^a^	1246	541	893.50 ± 352.50 ^ab^	1280.8	546.0	913.43 ± 367.43 ^ab^
Summer	2	285	282	283.50 ± 1.50 ^b^	1605	1382	1493.50 ± 111.50 ^a^	1660.8	1428.6	1544.72 ± 116.07 ^a^
Age		***			***			***		
7 yrs	129	298	252	270.51 ± 1.08 ^a^	1790	527	812.59 ± 28.93 ^a^	1800.3	548.8	830.22 ± 29.31 ^a^
8 yrs	198	296	240	268.40 ± 0.92 ^a^	1888	475	757.07 ± 21.10 ^a^	1902.0	484.4	774.23 ± 21.53 ^a^
9 yrs	57	274	241	256.04 ± 0.98 ^b^	693	446	579.44 ± 6.27 ^b^	698.7	508.3	598.32 ± 5.81 ^b^
Population		***			***			***		
Poba	40	282	250	261.50 ± 1.00 ^cd^	791	476	618.55 ± 12.85 ^bc^	809.6	525.1	638.89 ± 11.89 ^bc^
Godagari	70	286	253	268.21 ± 1.17 ^b^	987	527	698.31 ± 13.17 ^b^	999.9	548.8	712.97 ± 13.43 ^b^
Lalpur	39	298	252	279.54 ± 1.88 ^a^	1812	525	1046.33 ± 66.32 ^a^	1821.8	537.5	1076.13 ± 67.97 ^a^
Madargonj	70	296	252	277.21 ± 1.61 ^a^	1888	539	991.54 ± 48.38 ^a^	1902	550.3	1007.17 ± 48.99 ^a^
Jamalpur	40	295	253	269.53 ± 1.95 ^b^	1570	475	749.34 ± 40.71 ^b^	1584.4	484.4	767.57 ± 41.97 ^b^
Trishal	25	276	252	266.04 ± 1.37 ^bc^	790	538	651.28 ± 16.05 ^bc^	803.8	561.5	664.68 ± 15.16 ^bc^
Bhola	100	274	240	256.60 ± 0.97 ^d^	693	446	576.59 ± 4.39 ^c^	698.7	508.3	592.44 ± 4.06 ^c^
AEZ		***			***			***		
FFA	284	298	250	271.04 ± 0.73 ^a^	1888	475	810.19 ± 19.07 ^a^	1902.0	484.4	828.36 ± 19.39 ^a^
CA	100	274	240	256.60 ± 0.98 ^b^	693	446	576.59 ± 4.37 ^b^	698.7	508.3	592.44 ± 4.05 ^b^
CI, months		**			***			***		
<15	176	298	252	275.32 ± 0.89 ^a^	1888	536	923.63 ± 26.76 ^a^	1902.0	546.0	942.59 ± 27.29 ^a^
≥15	208	288	240	260.47 ± 0.72 ^b^	1040	446	601.89 ± 6.02 ^b^	1087.6	484.4	618.28 ± 5.98 ^b^
DP, days		***			***			***		
<90	18	296	282	292.11 ± 1.01 ^a^	1888	1078	1649.22 ± 42.21 ^a^	1902.0	1096.6	1671.24 ± 41.48 ^a^
90–150	55	298	254	280.55 ± 1.44 ^b^	1790	541	1058.71 ± 44.70 ^b^	1793.7	546.00	1087.14 ± 45.94 ^b^
>150	311	288	240	263.49 ± 0.61 ^c^	1219	446	642.56 ± 6.13 ^c^	1275.7	484.40	657.96 ± 6.16 ^c^

^1^ CLP: calculated lactation period, CLY: calculated lactation yield, SOC: season of calving, yr: years, AEZ: agroecological zone, FFA: Flood Fed Area, CA: Coastal Area, CI: calving interval, DP: dry period. ^2^ N: number of observations, Max: maximum, Min: minimum, SEM: standard error of the means. ^3^ NS: Non-significance, with “*” indicating significant at (*p* < 0.05), with “**” indicating significant at (*p* < 0.01), and with “***” indicating highly significant at (*p* < 0.001). ^4^ Different superscript letters indicate significant difference (*p* < 0.05) or highly significant differences among different levels of a certain factor.

**Table 2 animals-14-01248-t002:** Least-squares mean (LSM) of factors affecting the calving interval (CI) and dry period (DP) of indigenous river buffalo populations of Bangladesh.

Factors ^1^	N ^2^	Calving Interval, Days			Dry Period, Days		
		Max	Min	Mean ± SEM ^4^	Max	Min	Mean ± SEM ^4^
Overall	384	547	369	453.06 ± 5.64	303	74	185.78 ± 2.55
Parity		NS ^3^			NS		
3rd	172	537	370	452.40 ± 3.09 ^a^	291	75	185.18 ± 3.82 ^a^
4th	212	547	369	453.59 ± 2.72 ^a^	303	74	186.26 ± 3.42 ^a^
SOC		**			**		
Rainy	185	547	369	467.21 ± 3.36 ^a^	303	74	201.73 ± 4.18 ^a^
Autumn	195	485	369	440.91 ± 1.95 ^b^	238	76	172.10 ± 2.59 ^b^
Winter	2	420	392	406.00 ± 14.00 ^b^	140	136	138.00 ± 1.99 ^b^
Summer	2	379	370	374.50 ± 4.49 ^b^	94	88	91.00 ± 3.00 ^b^
LMP-300d, Kg		***			**		
<600	92	547	392	476.80 ± 3.54 ^a^	303	123	220.58 ± 3.81 ^a^
600–1000	240	535	395	457.53 ± 1.97 ^b^	286	110	190.66 ± 2.35 ^b^
>1000	52	464	369	390.40 ± 2.96 ^c^	184	74	101.65 ± 3.37 ^c^
Population		***			***		
Poba	40	480	397	456.30 ± 2.86 ^b^	227	141	194.80 ± 3.89 ^b^
Godagari	70	480	396	442.47 ± 2.29 ^bc^	218	110	174.25 ± 2.88 ^c^
Lalpur	39	480	369	415.13 ± 5.82 ^e^	209	74	135.58 ± 6.91 ^e^
Madargonj	70	480	370	424.94 ± 3.81 ^de^	227	74	147.73 ± 5.15 ^de^
Jamalpur	40	465	375	435.90 ± 4.02 ^cd^	212	80	166.36 ± 5.31 ^cd^
Trisal	25	465	410	445.44 ± 3.01 ^bc^	205	140	179.40 ± 3.8 ^bc^
Bhola	100	547	462	502.40 ± 2.15 ^a^	303	191	245.80 ± 2.41 ^a^
AEZ		***			***		
FFA	284	480	369	435.68 ± 1.71 ^b^	227	74	164.64 ± 2.26 ^b^
CA	100	547	462	502.40 ± 2.15 ^a^	303	191	245.80 ± 2.41 ^a^

^1^ SOC: season of calving, LMP-300d: level of milk production in 300 days, AEZ: agroecological zone, FFA: Flood Fed Area, CA: Coastal Area. ^2^ N: number of observations, Max: maximum, Min: minimum, SEM: standard error of the means. ^3^ NS: non-significance, with “**” indicating significant at (*p* < 0.01), and with “***” indicating highly significant at (*p* < 0.001). ^4^ Different superscript letters indicate significant difference (*p* < 0.05) or highly significant difference among different levels of a certain factor.

**Table 3 animals-14-01248-t003:** Descriptive statistics of test day (TD) milk yield in overall population.

TD	Milking Day	Number of Data	Mean (kg)	SDM ^1^	CV% ^2^	Minimum Milk Yield (kg)	Maximum Milk Yield (kg)
TD1	14th	384	1.51	1.04	68.85	0.50	6.00
TD2	42th	384	2.59	1.35	52.29	1.00	8.00
TD3	70th	384	3.66	1.29	35.25	1.50	9.50
TD4	98th	384	4.47	0.85	19.01	2.80	9.00
TD5	126th	384	4.11	0.85	20.75	3.00	8.13
TD6	154th	384	3.61	0.86	23.90	2.50	8.00
TD7	182th	384	3.08	1.04	33.62	1.30	7.45
TD8	210th	384	2.23	1.29	57.71	0.50	7.50
TD9	238th	380	1.58	1.18	74.32	0.50	6.10
TD10	266th	317	1.15	1.04	90.57	0.50	4.80
TD11	280th	86	1.77	0.94	53.19	0.50	4.00

^1^ SDM: standard deviation of the means. ^2^ CV: coefficient of variation.

**Table 4 animals-14-01248-t004:** Correlation analysis between production trait and reproduction trait.

	CLP	CLY	MY300d	CI
CLY	0.77 ***			
MY300d	0.76 ***	0.98 ***		
CI	−0.67 **	−0.73 ***	−0.73 ***	
DP	−0.81 ***	−0.79 ***	−0.78 ***	0.98 ***

CLP: calculated lactation period, CLY: calculated lactation yield, MY300d: milk yield in 300 days, CI: calving interval, DP: dry period. “**” indicates significant at (*p* < 0.01) and “***” indicates highly significant at (*p* < 0.001).

**Table 5 animals-14-01248-t005:** Parameter estimates and goodness of fit for the Wood model of test day milk yield of indigenous buffalo populations of Bangladesh.

Population	R²	RMSE	a	b	c	*t* _m_	*y* _m_
Poba	0.498	0.463	0.687	0.500	0.009	56.878	3.138
Godagari	0.559	0.425	0.679	0.525	0.009	59.869	3.446
Lalpur	0.396	0.512	0.937	0.465	0.006	73.084	4.324
Madargonj	0.411	0.527	0.903	0.490	0.008	64.681	4.269
Jamalpur	0.467	0.497	0.659	0.518	0.008	65.050	3.418
Trishal	0.659	0.436	0.386	0.677	0.010	64.993	3.307
Bhola	0.631	0.423	0.394	0.614	0.009	68.439	2.850

R^2^: Coefficient of determination; RMSE: root mean square error; a: lactation potential of the buffalo; b: rate of decline in the lactation curve; c: rate at which the curve reaches its peak; *t*_m_: peak lactation day (day); *y*_m_: peak milk yield (kg).

## Data Availability

The datasets generated and analyzed during the current study are available from the corresponding author on reasonable request.

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
