# Peer review of "Factors Affecting the Milk Production Traits and Lactation Curve of the Indigenous River Buffalo Populations in Bangladesh"

_animals, 2024, doi:10.3390/ani14081248_

Round 1
Reviewer 1 Report
Comments and Suggestions for Authors
Dear Author(s),
Please provide the revisions in the attached file.
Best Regards.

Reviewer 2 Report
Comments and Suggestions for Authors
Dear Authors,
You deserve praise for your efforts. Determining the duration and efficacy of lactation in Bangladesh buffaloes will contribute significantly to the breeding of buffalo. My recommendations are listed below.
1-The abbreviation "viz." is frequently mentioned in the article. I recommend utilizing alternative conjunctions in place of the current conjunction.
2-Additionally, "lactation period" should be substituted for "parity," which is utilized extensively in the article.
3-In line 134: It is incorrect to include references as "reference 4" within the midst of a phrase. The sentence should be rephrased by incorporating the name of the author.
Reviewer 3 Report
Comments and Suggestions for Authors
Review of manuscript no. animals-2951086, entitled Factors Affecting the Milk Production Traits and Lactation Curve of the Indigenous River Buffalo Populations in Bangladesh
Manuscript no. animals-2951086 contains quite interesting topic. As authors wrote, cattle and buffalo are the most important sources of milk and meat in Bangladesh. The understanding the impact of the environmental factors can help farmers from Bangladesh to manage dairy buffaloes herd. This manuscript needs major revision before I could consider to recommend that paper for publication.
Introduction
First of all the introduction section is too short, and mostly concentrate on buffalo population. The information about environmental factors that influenced milk production and reproduction should be extended. Also part concerned on lactation curves should be extended.
Material and methods
Lines 99-104 – Should be carefully checked, and degree Celsius symbol should be used instead zero (0) or o-letter (o) in superscript.
Line 142 – The Table S1 should be omitted and not presented as supplementary table.
Line 146 – Write “100 g” instead “100g”.
Lines 153-154 – The information about Excel spread should be omitted. It is obvious that data must be collected in a file and Excel file is one of the most popular options.
Lines 166-167 – R studio should be cited as: R Core Team (2019). R: A language and environment for statistical computing. R Foundation for Statistical Computing, Vienna, Austria. URL https://www.R-project.org/. Also all R packages should be cited and the citation of Agricolae in R-package is: Felipe de Mendiburu and Muhammad Yaseen (2020). agricolae: Statistical Procedures for Agricultural Research. R package version 1.4.0, https://myaseen208.github.io/agricolae/https://cran.r-project.org/package=agricolae.
Line 191 – There is no L effect in model 1 (see line 173), so write “(where n=1, 2 and 3, in which 1= less than 600 kg, 2 = 600-1000 kg, and 3= more than 1000 kg) for Model 2” instead “(where n=1, 2 and 3, in which 1= less than 600 kg, 2 = 600-1000 kg, and 3= more than 1000 kg) for Model 1 and Model 2” or correct the model in line 173.
Lines 194-198 – In results section there is lack of information about regression analysis. Please, add information about regression in result section or remove this lines from manuscript. Additionally, the information what correlation coefficients (Pearson or Sperman) were calculated should be added to this part of manuscript.
Line 203 – The formula of Wood function should be added and the estimated parameters (a, b and c) should be explained as under Table S3.
Lines 213-214 – The citation of nlme in R-package is required: Pinheiro J, Bates D, R Core Team (2023). nlme: Linear and Nonlinear Mixed Effects Models. R package version 3.1-164, https://CRAN.R-project.org/package=nlme.
Results
Tables 1 and 2 – Change the title of tables. Don’t write: “ANOVA and least square mean….”. The table should be on one page. Clarify in the header that in table are LSM. Unify the space before and after ± symbols (see winter and summer for CLY and milk yield of 300 days). Check the significance of difference between seasons for CLP. The difference are significant, and all four seasons had superscript “a”.
Lines 229-230 and 273-274 – If I proper understand, the values that differ significantly or highly significantly for a given factor (for example lactation) bear different letters. Please change the information.
Lines 231-232 – The sentence is not clear. Probably, the SOC, Age, Population, AEZ, CI, and DP were significantly affected the CLP.
Line 233 – In all text only means should be written instead means with SEM, which will improve the readability of the text (applies to whole manuscript).
Figure 3 – Combine Figure 3 and S1 in article as a graph composed of two: a) Average milk production and b) maximum milk production on test day records for different buffalo populations of Bangladesh under village condition.
Line 276 – The Table S2 should be moved from supplementary materials to manuscript.
Lines 284-286 – Rewrite this sentence.
Line 290 – Check the MP-300d abbreviation.
Table 3 – The table and footnote should be on one page.
Line 301 – In Table 3 there is no non-significant or significant correlation, so the sentence should be shortened: “**” indicating significant at (P< 0.01), and with “***” indicating highly significant at (P< 0.001).
Line 307 – The Table S3 should be moved from supplementary materials to manuscript as Table 4. This table contain goodness of fit parameters (R2 and RMSE) which are important when the lactation curves are modelled.
Discussion
Line 332 – If “SOC significantly affected (P>0.01) LY” you should change “>” on “<”.
Lines 360-361 – The sentence is not clear and should be rewritten. Probably CI was significantly affected by LMP-300d, population and agro-ecological zone (AEZ).
Line 381 – The TD milk production data are presented in Table S2 not S1.
Supplementary materials
Figure S1, Tables S2 and S3 as I wrote before should be moved to manuscript. Table S1 should be deleted. This way, important information will be included in the manuscript, and the supplementary materials part will be eliminated.
Table S2 – The values that differ significantly are in one column not in one row. Use only SDM or SEM in table. Write “Mean2 (kg)” in the header.
Round 2
Reviewer 1 Report
Comments and Suggestions for Authors
Dear author(s),
You can find my comments about your peer-reviewed manuscript, which ID is “animals-2951086”, in the Animals. I examined your manuscript again within the scope of the your revised manuscript file and I will take some notes from section to section with the line number, separately.
Best Regards.

Reviewer 3 Report
Comments and Suggestions for Authors
I accept the manuscript in present form.
Author Response
Thank you for your valuable comments.
Round 3
Reviewer 1 Report
Comments and Suggestions for Authors
Dear Author(s),
I reported my suggestion to the Editor.
Best regards.